# Applications of Silk Fibroin in Human and Veterinary Medicine

**DOI:** 10.3390/ma16227128

**Published:** 2023-11-11

**Authors:** Piotr Koczoń, Alicja Dąbrowska, Ewa Laskowska, Małgorzata Łabuz, Katarzyna Maj, Jakub Masztakowski, Bartłomiej J. Bartyzel, Andrzej Bryś, Joanna Bryś, Eliza Gruczyńska-Sękowska

**Affiliations:** 1Department of Chemistry, Institute of Food Sciences, Warsaw University of Life Sciences, 159C, Nowoursynowska St., 02-776 Warsaw, Poland; piotr_koczon@sggw.edu.pl (P.K.); joanna_brys@sggw.edu.pl (J.B.); 2The Scientific Society of Veterinary Medicine Students, Warsaw University of Life Sciences, 159, Nowoursynowska St., 02-776 Warsaw, Poland; s212294@sggw.edu.pl (A.D.); s208919@sggw.edu.pl (E.L.); s208920@sggw.edu.pl (M.Ł.); s195553@sggw.edu.pl (K.M.); s206880@sggw.edu.pl (J.M.); 3Department of Morphological Sciences, Institute of Veterinary Medicine, Warsaw University of Life Sciences, 159, Nowoursynowska St., 02-776 Warsaw, Poland; bartlomiej_bartyzel@sggw.edu.pl; 4Department of Fundamental Engineering and Energetics, Institute of Mechanical Engineering, Warsaw University of Life Sciences, 164, Nowoursynowska St., 02-787 Warsaw, Poland; andrzej_brys@sggw.edu.pl

**Keywords:** silk, proteins, polymers, combinations, medicine, veterinary, cartilage regeneration, bone tissue regeneration, wound healing, vascular surgery

## Abstract

The properties of silk make it a promising material for medical applications, both in human and veterinary medicine. Its predominant amino acids, glycine and alanine, exhibit low chemical reactivity, reducing the risk of graft rejection, a notable advantage over most synthetic polymers. Hence, silk is increasingly used as a material for 3D printing in biomedicine. It can be used to build cell scaffolding with the desired cytocompatibility and biodegradability. In combination with gelatine, silk can be used in the treatment of arthritis, and as a hydrogel, to regenerate chondrocytes and mesenchymal cells. When combined with gelatine and collagen, it can also make skin grafts and regenerate the integumentary system. In the treatment of bone tissue, it can be used in combination with polylactic acid and hydroxyapatite to produce bone clips having good mechanical properties and high immunological tolerance. Furthermore, silk can provide a good microenvironment for the proliferation of bone marrow stem cells. Moreover, research is underway to produce artificial blood vessels using silk in combination with glycidyl methacrylate. Silk vascular grafts have demonstrated a high degree of patency and a satisfactory degree of endothelial cells coverage.

## 1. Introduction

Silk is a protein polymer of natural origin that is produced by certain arthropods such as spiders, mites, fleas and silkworms [1]. However, the structure, composition and properties of silk extracted from different animals are not the same, but vary according to function and origin [2]. So far, in terms of spinning and details of the structure, silk, which is produced by silkworms, has been studied the most deeply among the above-mentioned silk types [3].

There are two main types of silk fibres, which are classified according to the species of the silkworms from which the type is extracted [4]. These are mulberry silk (extracted from the mulberry silkworm *Bombyx mori*) and tussock silk (extracted from the oak silkworm *Atheraea pernyi*). There are two more types of silk available on the market, namely Eri silk (extracted from the Eri silkworm, only in India) and Muga silk (from Muga silkworm). However, this review will focus on the applications of silk obtained from the mulberry silkworms *Bombyx mori*, because this is the type which has found the widest application in medical sciences.

Silk itself is composed of two main components, namely silk fibroin (SF) and sericin [5]. For the medical applications in transplantology, sericin as a component of silk is unnecessary and can be even problematic, because its presence worsens the biocompatibility of grafts made using this material [1]. Therefore, it is crucial to remove residual sericin from silk that would be used for that purpose [6]. Such a process is called degumming, and it involves subjecting silk to thermal and chemical reactions that lead to the removal of sericin [5]. SF itself shows excellent tissue compatibility [5], even better than synthetic polymers, which will be discussed in later parts of this review. However, the presence of sericin will not always be unfavourable. The advantage of sericin is, e.g., antibacterial behaviour against *E. coli* or *S. aureus* [7], which can be also successfully applied in therapy of, for example, skin diseases.

Thanks to its mechanical properties, silk has found extensive use in the biomedical sciences. The production of silk implants, for example, is possible because of the specification of this material, which is exceptionally favourable when considering its interaction with the organism of the potential recipient. Studies have shown that it has excellent cytocompatibility, which accounts for its potential use in tissue engineering [8]. Transplant materials must be non-allergenic, non-toxic, biodegradable and biocompatible [4]. Silk, which exists as the core of a structural protein fibre coated with already-mentioned sericin, is a highly biocompatible material [8], provided it is properly processed, and that means subjected to degumming. Silk therefore fulfils the main requirements set for transplant materials; in particular, it is biocompatible, biodegradable and has a low immunogenicity [9,10,11,12,13,14]. Most of the amino acids that make up silk fibre are glycine (45.9%) and alanine (30.3%), both of which show minimal chemical reactivity, so that grafts made from this fibre very rarely cause a rejection reaction [15,16]. In addition, silk and the grafts made using SF are porous structures, which simplifies the cell proliferation at the transplant site [5,17]. Another characteristic of a good transplant material is its resistance to various physical conditions, such as the high temperature of the sterilisation process. Even under extreme conditions the material must not lose its natural properties. Silk fibres have high ductility and breaking strength, which are also assets in biomedical applications [4]. Its fibres are very strong, but flexible [18].

Interestingly, SF is a material that can be used for 3D printing, a method that is increasingly being used in biomedical sciences. The growing prominence of 3D printing technology in the realms of human and veterinary medicine, spanning applications in bone, skin, cartilage, and other tissue regeneration, underscores its escalating significance in advancing therapeutic solutions [19,20,21].

The integration of SF and 3D printing technology showcases a promising approach to stimulate chondrogenesis and osteogenesis [21]. In the realm of wound healing and regenerative medicine, innovative techniques are continually emerging to enhance the treatment of skin damage. Through the incorporation of bioengineered materials, such as SF and gelatine hydrogel, researchers are actively pursuing more effective wound healing solutions for both human and veterinary medicine. These advancements in bioengineered hydrogels and 3D printing techniques play a pivotal role in diversifying the range of therapeutic materials available for wound care, with the ultimate goal of accelerating the healing process and improving patient outcomes [19,22].

Because of its structural proteins, silk plays an increasingly important role in human and veterinary medicine. It is used, for example, in assisting the healing process of wounds, as discussed further in this paper. It therefore possesses a number of characteristics that account for its potential as a therapeutic agent. It is characterised by its easy availability and, moreover, it is not expensive to obtain [9,23,24]. However, it should be borne in mind that silk threads not subjected to any mechanical, chemical or genetic treatment are of little value in medicine, as they lack bioactive properties [9,25]. The situation is improved by the structure of the material, which facilitates its bonding with other substances supporting therapy.

The main advantages of silk fibroin are presented graphically in Figure 1.

This review paper aims to present and discuss applications of silk combinations with selected proteins and artificial polymers in human and veterinary medicine. Although SF usage for tissue regeneration is currently a widely studied issue, articles describing its application in veterinary medicine, especially new ones, are still quite difficult to come by. Our hope is that outlining the status quo by presenting some of the recent discoveries in this field will broaden the overall knowledge on this subject and make it easier to comprehend or maybe even encourage some researchers to study it deeply and provide new solutions that will improve tissue engineering in veterinary patients.

## 2. The Use of Silk in Cartilage Regeneration

Cartilaginous tissue is a type of connective tissue that creates structures such as the nasal septum or auricle, provides scaffolding for trachea and bronchi, but most of all, plays an important role in forming bone articulations [26]. The cartilage covers their connection points, taking a part in forming joints and therefore alleviating some movement-related overloads [27]. Because of not being vascularised nor innervated, cartilage has very limited regeneration potential and so the continuous friction and pressure lead, over time, to its wear and damage. SF may find its use in therapy of exactly such damages [28,29].

For regenerative therapy of cartilaginous tissue, many different forms of SF scaffolds can be used—mats, films, hydrogels, fibres, sponges or “bioinks” for 3D printing [30]. In general, all aforementioned forms of this polymer increase secretion of intercellular matrix of the cartilage as well as provide a basis for chondrocytes to grow and proliferate [29]. This last process can be additionally improved by using growth factors (GF), such as Rb1 or TGF-β1, built into the SF scaffolding [28,29]. Activity of GF, for example above-mentioned TGF-β1, can be regulated by using nanoparticles of chitosan—a polysaccharide derivative of chitin—during SF scaffold formation. This solution has been tested in vitro and in vivo in regenerating rabbits’ knee joint cartilage [31]. Things are similar when it comes to bone tissue regeneration, but more on that in the next chapter.

Creating scaffolds for rebuilding cartilaginous tissue is not the only medical use of SF. It has been shown that SF can also encourage canine adipose-derived multipotent mesenchymal stromal cells (cADMMSC) differentiation into chondrocytes. When cultivated in a standard cell culture medium, on a basic polystyrene surface, cADMMSC have the ability to differentiate into chondrocytes, osteocytes or adipocytes. However, when grown in a standard medium on a surface of SF film and scaffold, despite not using any specific chondrocyte medium, the cells strongly preferred differentiating into chondrocytes [32]. Thus, the method offers the possibility of using SF to repair cartilage injuries with patient’s own adipose-derived stem cells.

Moreover, recent studies have shown that SF scaffolds enriched with kartogenin (KGN), a small-molecule compound, reported to encourage chondrocyte differentiation from bone marrow-derived stem cells [33] and to exhibit anti-inflammatory properties [34] were able to promote cartilage tissue regeneration in rats. In this new study, SF served as a scaffolding for KGN-containing liposomes, allowed its slow release and provided mechanical support for the healing cartilage defect, while KGN enhanced extracellular matrix secretion and chondrocyte proliferation, as well as alleviated oxidative stress in the damaged area which, combined, resulted in an increased cartilage restoration score (both in micro- and macroscopic observation), compared to the control group [35]. Therefore, the usage of SF and KGN scaffolds may probably be a way to reduce negative effects of oxidative stress in cartilage injuries and disorders and make healing easier and faster for both the animal and human patients.

Another way of repairing cartilaginous tissue with SF is using a hydrogel with built-in particles of etanercept [36], an immunosuppressive agent, used in rheumatoid arthritis treatment in humans, tested on animal models [37]. To investigate this solution, a dedicated hydrogel containing SF and pullulan (an exopolisaccharyde produced by *Aureobasidium pullulans* yeast [38]), with addition of etanercept, has been prepared. The hydrogel was incubated in an appropriate buffer and implanted into an osteochondral defect in a rabbit knee joint, which has been made beforehand under laboratory conditions. After 8 weeks, the injuries healed completely and correctly in all test animals, which potentially gives a chance of treating cartilage defects in animals and humans in an innovative way, reducing the body’s immunological response to exogenous therapeutic compounds [36].

Similar results have been obtained using a composite scaffold of collagen and SF embedded with poly(lactic-co-glycolic acid) (PLGA) microspheres containing TGF-β1 [39]. The combination of SF with a natural polymer (collagen) and synthetic but highly biocompatible nanocarriers have encouraged bone marrow stem cells to differentiate into chondrocytes and thus stimulated cartilage repair in the rabbit knee joint [40]. This suggests that, in the future, such scaffolds could potentially help regenerate cartilage defects in human and animals, without causing unnecessary damage via more invasive methods and scarring.

It is also possible to use 3D printing methods for the regeneration of cartilage tissue. The study on developing regenerative implants inducing osteogenesis and chondrogenesis, addressing hip dysplasia, highlights the valuable attributes of SF. The innovative implant, manufactured using advanced 3D printing techniques, incorporates SF to actively stimulate chondrogenesis. Notably, the implant exhibits strong biocompatibility, as verified through various assessments. Through the integration of 3D printing and SF, this regenerative implant effectively triggers osteogenic and chondrogenic responses, ultimately restoring the natural form of the acetabulum. Consequently, it offers a promising avenue for the treatment of hip dysplasia [21].

It is worth mentioning that, apart from regenerating cartilaginous tissue with SF scaffoldings, its extracellular matrix (ECM), previously separated from the cells, can be used, combined with SF, to repair injured bones. This solution is possible because bone formation occurs mostly on a cartilaginous base (endochondral ossification) [26]. Fibroin scaffold, enriched with cartilaginous tissue ECM, although for the moment only in in vitro studies, may be used as a foundation for bone tissue regeneration [41], along with other methods described further.

Forms of SF applied in cartilage regeneration are presented graphically in Figure 2.

In summary, SF itself is an excellent scaffold for regenerating cartilage; however, combining or embedding it with different substances might have beneficial effect on the healing tissue. For example, by adding TGF-β1 to the scaffolding chondrocyte proliferation can be increased, which is quite desirable in a sparsely vascularised tissue. Enriching it with kartogenin may support the healing process by reducing oxidative stress, which has destructive impact on cartilage tissue and is still an actual problem, and combining SF with etanercept allows it to serve its purpose without causing potential harm via inflammatory reaction. Finally, SF scaffoldings by themselves can potentially drive patients’ adipose-derived stem cells to differentiate into chondrocytes, which reduces the problem of finding a donor, and combined with collagen, embedded with PLGA/TGF-β1 microspheres, can cause bone marrow stem cells to differentiate into chondrocytes and proliferate to restore normal structure of the cartilage, without leaving any scars that would disrupt its proper functioning.

## 3. The Use of Silk in Bone Tissue Regeneration

Bone tissue, like cartilage, is classified as connective tissue. It is composed of both mineral and organic substances. Bones have many key functions in the body—they store important elements, produce the bone marrow and mechanically protect internal organs. They also play an important role, together with the already mentioned cartilage tissue, in locomotion of the body [42]. Silk, extracted from the cocoons of *Bombyx mori* [8], has also become the focus of research into the treatment of bone damage. Developing the available knowledge on such therapies is extremely valuable, as bone damage is among some of the most common injuries with which a patient is referred to the operating theatre. The prevalence of this type of pathology, and the need to expand the available literature in this area, is evidenced by the sheer development that animal orthopaedics has experienced in recent years.

The desire to recreate the structure of bone requires consideration of its composition and the role it plays in the body. Investigating the use of silk in the treatment of bone tissue damage was therefore focused on how to balance the bioactivity with the mechanical properties of the printed bones [43]. A number of compounds, such as tricalcium α-phosphate (α-TCP) or hydroxyapatite (HA) [44,45,46], have been used to make bioplastics for cell scaffolds, but, interestingly, it has been shown over time that adding silk to these compounds improved the properties of the final product [43]. The structure of some biomaterials allows the generation of special 3D fibrous structures, which are supposed to mimic the natural ECM. The generation of such structures is made possible, for example, by the electrospinning method, which allows fibres to be produced in different spatial orientations and enables cell culture proliferation to be controlled [44]. It is a versatile technique that enables the production of nanofibre-based biomaterial scaffolds, which then mimic the nanoscale properties of certain ECM components in tissues [47]. In short, then, it is a method of producing micro-scaffolds, enabling control of the mechanical properties of the resulting fibres that build such scaffolds [44]. Silk is a polymer that is perfectly suitable for the production of such structures [48]. Regarding bone tissue, it is also possible to add bioactive factors and stem cells to the silk matrix to create an osteogenesis-friendly environment, which can additionally be controlled in various ways to direct cell regeneration [49], similar to what was discussed for cartilage. There are studies available confirming that the combination of SF, polyhydroxybutyrate-co-(3-hydroxyvalerate) (PHBV) and HA can provide a favourable environment for cell proliferation during bone regeneration. The role of PHBV and silk was to create conditions for cell adhesion, proliferation and differentiation, while HA was responsible for the induction of ECM production. Subsequent analyses (scanning electron microscopy combined with energy dispersive X-ray analysis (SEM/EDX), Fourier transform infrared spectroscopy (FT-IR), mechanical uniaxial tensile and compression tests, biodegradation tests and in vitro bioactivity tests) showed a smooth and homogeneous fibre arrangement and gradual biodegradation of the components used. Despite the disappearance of the fibrous components of the microenvironment, the fibrous structure of the sample itself was preserved. At the same time, preliminary in vitro bioassessment showed that after 1 and 3 days of culture, the cells adhered to the fibres, preserving their morphology while presenting a flattened appearance and elongated shape on the fibre surface [50].

In the new research, SF and HA were also combined with titanium oxide (TiO_2_) to produce a bone-mimicking scaffold to support bone tissue formation. The scaffolds were cultured in vitro with MC3T3, i.e., osteoblast precursors obtained from the calvaria of mice *mus musculus*. The results demonstrated that the scaffolds with HA and TiO_2_ exhibited very satisfying biological performance: cell adhesion, viability, proliferation, alkaline phosphatase activity and calcium content better than the scaffolds without HA and TiO_2_. Moreover, they supported the synthesis of calcium secreted by cells [51].

Another, slightly earlier study also showed that a 3D porous composite scaffold obtained from biodegradable SF scaffold doped with mineralised collagen (MC) could induce bone regeneration in cranial defects in rats. The results showed that scaffolds could promote the regeneration of both new bone and the vascular system. This combination exhibited good mechanical properties and the ability to regulate the degradation rate of the scaffold. All of this provided favourable conditions for the proliferation of bone marrow mesenchymal stem cells (BMSC) and preosteoblasts (MC3T3-E1). An additional advantage is the relatively inexpensive cost of producing such scaffolds [52].

A three-component bioplastic consisting of polylactic acid, hydroxyapatite and silk was also used to produce bone clips, which had good biocompatibility and satisfactory mechanical properties [53].

Another method is the possibility of low-temperature print using a mixture composed of collagen, decellularised extracellular material and SF. The product of such printing also creates good conditions for cell proliferation and differentiation [43].

The study, which employs 3D printing technology to fabricate inventive scaffolds composed of SF, collagen and HA (SCH) infused with recombinant human erythropoietin (rh-EPO), reveals that these composite scaffolds exhibit gradual degradation, foster the proliferation of osteoblasts, and notably improve the reconstruction of bone defects, offering substantial potential for clinical applications. The use of a low-temperature 3D printing method effectively maintains the biological activity of SF and collagen, resulting in scaffolds with optimal properties [20].

Another form in which silk can be used to be effective in the treatment of bone injuries are hydrogels. Their microstructure closely resembles the structure of the tissue, which is a major advantage. For bone defects treated with hydrogels containing silk, therapeutic effects were achieved in a short time. In addition, the healing process was accompanied by other positive elements, such as increasing the thickness of the bone trabeculae or increasing the rate of mineral deposition. Nanohydroxyapatite was also added to the hydrogel, which further improved cell metabolic activity and naturally resulted in more efficient bone regeneration. Due to their biocompatibility and low immunogenicity, injectable silk hydrogels, on the other hand, can provide an ideal carrier for therapeutic agents [49]. For example, a hydrogel has been developed that, when administered, simultaneously released bioactive magnesium, silicon and strontium ions, creating a microenvironment suitable for osteo- and angiogenesis [54]. In addition, studies have also been conducted showing that the SF hydrogel can be combined with GFs, such as VEGF or BMP-2. Administration of GFs alone may result in a number of undesirable effects, such as the formation of ectopic bones. Such administration of isolated GF may also result in therapy failure, due to the short half-life of these proteins [55]. So, in practice, for the effective carriage of BMP-2, SF hydrogel was fabricated using the electrospun poly(ε-caprolactone) (PCL) nanofibre mesh tubing. The results demonstrated that this hydrogel was very effective carrier for BMP-2, and this combination resulted in promising level of bone formation. Moreover, the silk hydrogel has been completely degraded, and the repair of large bone defects in that case was satisfying [56].

SF hydrogels were also used in a study on lacunar bone regeneration. For this purpose, SF was combined with mesoporous bioglass and sodium alginate. By activation of the mitogen-activated protein kinase (MAPK) signalling pathway, the hydrogel caused an osteogenesis-friendly environment [57].

It is also possible to prepare a special silk nanofibre film using electrospinning, which is then able to mimic the microstructure of the natural ECM. A study conducted on a rabbit calvaria defect model has demonstrated new bone formation in vivo, and moreover it was found that cells on this type of substrate presented higher alkaline phosphatase activity and more efficient osteocalcin production [49]. In the production of this type of film, it is also possible to mix various polymers with SF to improve the stability and mechanical properties of the final product. A promising blend is the combination of silk with chitosan, as the properties of both polymers enable the regulation of cytokine and GF secretion in the defect and the stimulation of cell proliferation and differentiation. Interestingly, this type of combination can be used not only as an implant shell to repair bone damage, but also as a scaffold in tissue engineering to treat defects of the skin, cornea, adipose tissue or other soft tissues. Another study [58] compared the proliferation and differentiation capacity of rat bone marrow-derived mesenchymal stem cells into bone and adipose tissue cells on silk-chitosan film and on polystyrene tissue culture plates, and the results have proved that a thin film of SF and chitosan not only provided a comparable environment for growth and proliferation, but also promoted the differentiation of stem cells into bone and adipose tissue cells.

Chitosan has also been combined with SF in other, more recent studies. Two-layer multifunctional nanofibre mats were successfully produced in some research—the osteogenic side of which consists of SF/PCL and the antibacterial side of PCL/chitosan). The results demonstrated that such mats were characterised by very favourable biocompatible properties, sufficient hydrophilicity and appropriate mechanical properties, as well as high tear strength [59]. The antibacterial aspect of such mats seems to be particularly beneficial here, because this property can significantly reduce postoperative bacterial infections caused by grafts obtained in that way.

Forms of SF applied in bone tissue regeneration are presented graphically in Figure 3.

## 4. The Use of Silk in Wound Healing and Skin Regeneration

Various types of wounds and skin damages are the next extraordinarily prevalent issue in both human and veterinary medicine. The therapy of such afflictions and injuries is often a challenge, significantly impacting the quality of life for patients. This exerts pressure upon scientists and doctors to combat this issue with utmost efficacy [30]. Both acute and chronic conditions are challenging not only for physicians, but also for patients and animal owners, primarily due to the difficulty in providing an appropriate wound healing conducive environment. All forms of skin damage represent a potential threat, as this organ serves as the primary barrier between the external environment and the organism. Disruption of this barrier is an open gate for pathogens, which can be a lethal threat [10,60]. In animals, it is particularly challenging to maintain dressings, change them (if the animal is hyperactive, aggressive or wild) and prevent the patient from accessing the site of injury. It is therefore extremely important that the healing process proceeds quickly, without discomfort or complications and with the least possible interference from the veterinarian or owner. In addition, in both animals and humans, it is common to encounter wounds that are extensive, deep or difficult to heal due to a variety of factors, such as associated diseases, e.g., diabetes, or immunological deficiencies [9,12,13,24,61]. Such damages necessitate specialised treatment and materials that adequately safeguard the injury environment and provide a matrix for tissue regeneration, thereby facilitating wound closure [30].

To produce economically viable and thus widely accessible medical materials based on silk for wound treatment, two types of three-dimensional fibroin structures are employed: nanofibrous mats and micro-porous scaffolds. These structures are subsequently coated or combined with specific agents aimed at conferring bioactive characteristics [9]. Such agents can include other proteins secreted by silkworms, like sericin, antibiotics, anti-inflammatory factors, or agents promoting cell proliferation and migration [13,25,30].

All chemical, biological, or genetic modifications aim to achieve precisely defined, highly functional traits that underlie the therapeutic properties of silk-based materials. These modifications result in various types of dressings and fillers that create an environment around injuries and within wounds, facilitating and expediting the entire wound and skin damage healing process. Therapeutic materials used in treatment must exhibit low immunogenicity. SF has this characteristic, as well as many others that account for its therapeutic value. Through functionalisation using additional factors, scaffolds, mats, aerosols, fillers, films, or hydrogels produced from silk proteins, further desired medical attributes are instilled. These attributes include biocompatibility, biodegradability, versatility, and stability in both intra- and extracellular environments.

Genetic engineering is one of the methods of obtaining SF with specific characteristics. The process of gene recombination significantly facilitates and accelerates the production of functional 3D matrices from fibroin. Thanks to genetic engineering, the production of SF with the desired characteristics is much faster and more efficient. The most efficient method of producing functional fibres is based on the synthetic coding of the relevant amino acid sequences. Recombinations leading to silk fibre functionalisation can be achieved not only through alterations in the fibroin amino acid sequence, but also by adding functional proteins [23]. Appropriate variations allow control of potential therapeutic agents of a future matrix composed of recombinant silk proteins [13]. Bioengineering also enables the incorporation of non-therapeutic (but equally important) features, such as faster degradability or the ability to change conformation or a state of aggregation under specific stimuli [14], which has already been mentioned in the fragment about bone and cartilage therapy. These modifications contribute to enhanced biocompatibility, primarily due to an increased capacity for cell attraction and adhesion to the matrix [12,13,60]. Studies conducted on mice and rats have demonstrated that nanofibrous structures enriched with molecules that promote cell adhesion, such as integrins, not only facilitate cell adhesion but also promote cell proliferation, accelerate neovascularisation, and enhance tissue healing [13,60,62,63]. Another pivotal aspect of silk fibre functionalisation involves conferring antibacterial properties. Antibacterial factors hold immense importance in the preparation of wound dressings, graft matrices, or silk-based fillings, as they prevent infection development, thus providing a suitable environment for healing and significantly speeding up the recovery process [13,23,64]. The fibres forming therapeutic 3D structures can be enriched with antibiotics, as well as proteins like defensins or hepcidins [23]. Coating fibroin with antibacterial agents is particularly crucial because non-functionalised silk material in this regard could foster microbial colonisation within it, thereby precluding its therapeutic properties [13]. Another additive imparting desired traits to fibroin can be other molecules, both organic and inorganic, such as silaffin (which facilitates the deposition of silica on the biomaterial surface, promoting cell proliferation matrix formation and stabilisation) [65], gelatine or glycol, calcium ions, sodium chloride, and glucose (known as porogens, which support the formation of pores in nanofibrous scaffolds) [25,30,65].

There are numerous types of dressings produced by enriching silk nano- and microfibres, including mats, scaffolds, films, hydrogels, spheres, matrices, coverings, aerosols, and sponges [23,25]. They have a very wide range of applications, from uncomplicated traumatic wounds, post-operative wounds, deep loss of large skin areas, diabetic wounds, and burns, to advanced and extensive skin grafts. Comparative studies between functionalised silk dressings and other wound dressings have demonstrated that enriched fibroin reduces the expression of pro-inflammatory cytokines while increasing the expression of anti-inflammatory cytokines in burn cases [13]. Another study showed the positive impact of fibroin sponge dressings on the healing of challenging diabetic wounds [61]. The creation of effective hydrogels using SF represents a significant advancement in the field of wound repair. Renowned for its robust biocompatibility, SF serves as a fundamental building block for constructing these hydrogels. One of its key attributes is controlled biodegradability, allowing for precise customisation to meet specific tissue regeneration needs, such as promotion of vascular regeneration, which regulates the local immune response. These hydrogels benefit from SF’s excellent mechanical properties, enhancing their stability and overall efficacy. SF’s exceptional self-assembly capabilities play a pivotal role in developing hydrogels with remarkable stability, making them essential components in tissue engineering and regenerative medicine. This innovative approach opens new avenues for designing biocompatible, customizable hydrogels that hold great promise for various applications in the field of healthcare and biomedical research [66,67].

Additionally, some innovative techniques in tissue engineering and regenerative medicine such as the 3D printing highlight the ongoing efforts to enhance wound healing treatments. By incorporating bioengineered materials like SF and gelatine hydrogel into the development of artificial skin, researchers are actively working to create more effective wound healing solutions for both human and veterinary medicine. These advancements in bioengineered hydrogels and 3D printing techniques contribute to the diverse range of therapeutic materials available for wound care, ultimately accelerating the healing process and improving patient outcomes [19,22].

Silk has been used for centuries in the skin damage treatment, and today, there is an increasing focus on studying its therapeutic properties and actively seeking new technologies and applications. Research efforts are expanding not only in animal models, but also in human trials, and their results demonstrate the positive impact of utilising SF in treatments. These findings underscore silk’s potential as a promising material not only in medicine, but also in other fields such as the food and cosmetic industries [65]. In summary, the goal in wound healing for both human and veterinary medicine is to effectively accelerate the healing process, thereby enhancing patient comfort and treatment efficacy. Silk proteins serve as excellent material, and when functionalised, they acquire numerous characteristics conducive to healing even the most challenging skin damage. Various types of dressings are produced by enriching SF fibres with diverse factors that positively influence the recovery process. Additionally, its widespread availability and relatively low production costs are crucial features. This accessibility ensures that new technologies and therapeutic material production are feasible worldwide, bridging the gap between developed and developing countries.

Forms of SF applied in wound healing and skin regeneration are presented graphically in Figure 4.

## 5. The Use of Silk in Vascular Surgery

Another field of medicine where SF could find wider use is the treatment of damage of the cardiovascular system. Blood vessels are the way for oxygen and nutrients to be transported to the tissues and organs, so their damage can be a significant problem. Therefore, the development of methods of treating such injuries is as necessary as the previously discussed methods of treating defects in bones, cartilage or common body integument.

According to the available literature, many of the damage to the circulatory system can be eliminated by transplants using materials made of polymers, such as SF. However, not every polymer is suitable for such a graft, because there are several conditions that such material must meet before it can be implanted in the course of an artery or other vessel. Important aspects to pay attention to are the thrombogenicity and the patency index of the used materials. Thrombogenicity is the ability of a material to cause blood clot formation. The patency index means the assessment of the patency of the vessel after surgical intervention, taking into account the dimensions of its lumen, as well as the space sufficient for blood flow. This is especially important when you want to reconstruct small-diameter vessels, where it is much more difficult to ensure a sufficiently high patency index. A mixture of biocomponents constituting a hydrogel obtained from SF and glycidyl methacrylate (Sil-MA) was used to build vessels of this type. The resulting hydrogel had satisfactory mechanical and rheological properties, offering a perspective for the treatment of vascular injuries in the brain and ear [43], which are distinguished by their high anatomical complexity and the small size of their lumen. However, there is an ongoing requirement to optimise the hydrogel for thrombogenicity and to improve the patency index.

In other studies, it was indicated that melanin nanoparticles combined with SF may contribute to a change in the transparency of the printed material, so as to adapt poly(ethylene glycol) tetraacrylate to improve the resolution of printing blood vessels or empty test tubes, which contributes to improving the quality of the produced material [43]. SF obtained from the mulberry silkworm (*Bombyx mori*) was tested in combination with polyethylene terephthalate (PET) as a material for the graft of blood vessels with a diameter smaller than 6 mm. The test has been performed on 24 rats, and the material has been transplanted into the abdominal aorta with PET coated with SF(Glyc)—with glycine used as a porogen—(12 rats) and for comparison with PET coated with gelatine (12 rats), as the latter graft was already widespread in commercial use. In the graft with the use of SF(Glyc), more collagen fibres and more macrophages could be observed in the analysis of the histopathological image than in gelatine-coated PET. Additionally, it was noted that the number of macrophages in the case of SF decreased over time—the first microscopic examination took place 2 weeks after implantation, and another 3 months later. The SF coating was biodegraded, as evidenced by the migration of inflammatory cells and the promotion of site remodelling. Inside the graft, fibroblasts, the previously mentioned collagen fibres and blood vessels were observed. In addition, the presence of endothelial cells and smooth muscles was demonstrated throughout the central part of the implanted vessel. All these parameters confirm the ability of SF to remodel into native tissues of the body, and show that PET coated with SF(Glyc) can be used to transplant vessels with a diameter smaller than 6 mm [68].

Another use of SF in the treatment of cardiovascular diseases is the use of this material as a regenerative artificial material, which is possible due to its high biocompatibility. However, the time needed for the biodegradation of SF is long, therefore it is helpful to use the SVVYGLR peptide, which accelerates this process. Reducing the time needed for biodegradation makes it possible to prevent the occurrence of side effects of the transplant, such as an inflammatory reaction. The migration of epithelial cells as well as smooth muscle cells occurs faster, which contributes to a faster similarity of the graft to a natural blood vessel [69].

Another material combination that was tested for angiogenicity was SF with polyurethane and the SVVYGLR peptide, which was implanted into the abdominal aorta of rats. The study was designed to test the ability of fibroin-based materials to promote the growth of blood vessels. After the transplantation, histological preparations were performed, which confirmed the presence of endothelial cells, smooth muscles and fibroin. The cell layer formed was similar to the cellular composition of the original blood vessel. This is due to the similar amino acid sequence in the SVVYGLR peptide to the sequence that occurs in natural factors that promote angiogenicity. There were no inflammations, no calcifications and moderate biodegradability. To compare the ability of the SVVYGLR peptide to induce angiogenesis, a transplant was also performed using only SF with polyurethane. Despite similar cellular composition in both cases, more advanced angiogenesis was found in the presence of the SVVYGLR peptide [69].

Moreover, in the manufacture of vascular grafts, grafts made of electrospun SF were much more flexible than scaffolds made of synthetic polytetrafluoroethylene (ePTFE), which was considered the gold standard for this type of procedure. Vascular grafts made of silk closely matched the elasticity of primary vessels in rats. Furthermore, in vitro, electrospun SF showed a positive interaction with endothelial cells and blood components [48].

Another interesting study was published in 2023 on the role of a sirolimus-embedded silk microneedle (MN) wrap as an external vascular device for drug delivery efficacy. This study was conducted on dogs—a vein graft model was developed to interpose the carotid or femoral artery with the jugular or femoral vein. The results of this research show that sirolimus-embedded silk-MN wrap in a vein graft model has successfully delivered the drug to the intimal layer of the vein grafts. MN avoided vein graft dilatation, shear stress and decreasing wall tension and also it prevented to develop neoitimal hyperplasia [70].

Research using SF for the development of blood vessels therapy is also conducted for the treatment of coronary arteries. The desire to perform transplants for this type of arteries requires the creation of synthetic blood vessels, which, as the natural originals, will have a small cross-sectional diameter and which could be used, for example, during bypass implantation. SF derived from the silkworm *Bombyx mori* turned out to be a material that holds great promise for the reproduction of such blood vessels. It shows very good adaptability, which was mentioned earlier. Moreover, it has properties for efficient migration of endothelial cells and is able to control the local inflammatory response [71].

SF was also used in studies on the lymphatic system. The test group consisted of 68 patients undergoing kidney transplantation. In the first group of patients, lymphatic vessels were closed using the Enseal method, while in the remaining patients, the vessels were ligated with the use of silk. The results showed no clinically significant differences when using the above-mentioned methods, and there were no differences in complications between the bipolar method of closing lymphatic vessels and the method using silk [72].

Forms of SF applied in vascular surgery are presented graphically in Figure 5.

## 6. Challenges

Although SF is a promising material for use in biomedical sciences, studies also point to some drawbacks in its use. Among other things, some papers highlight the fact that the structure of silk appears to change at different stages of tissue regeneration. In particular, this refers to the loss of strength resulting from the altered crystallinity and reorientation of the β-sheets of the SF. The need to control the properties of the biomaterial during processing is therefore also highlighted, due to the complexity of its molecules. Standardisation of the process and production parameters and equipment also seems to be an important element when commercialising silk for that kind of purpose, and is seriously considered [4].

Regarding the strength aspect of grafts made using SF, their sensitivity to matrix metalloproteinases (MMPs) should also be mentioned. Among other things, this occurs because proteinase K has a high affinity for β-sheets component of SF. Collagenase is also able to degrade such material effectively. This aspect is further disadvantaged if one considers that various proteases are often overexpressed at the site of chronic wounds, which should always be kept in mind when wishing to introduce silk material as a therapeutic agent [24].

Attention is also drawn to the need for new additives to improve the printability and mechanical properties of photo-crosslinked prints made using SF. Many such additives may compromise the advantages of SF inks [15]. Other studies point to the need to optimise the rate of biodegradation of SF products to match the regeneration rate of new tissue [43]. However, despite such challenges, photo-crosslinked SF holds great promise for 3D printing [15].

Possible challenges can also be encountered when wishing to produce synthetic silk based on the genetic code. Although it is possible to design genes synthetically and to produce silk with desired properties, this method still needs some improvement. The main disadvantages of recombinant silk production are low yield of a particular silk protein variant, the problem of scaling up the process, and possible contamination with endotoxin [23].

From a veterinary point of view, there are only a few investigations demonstrating the use of SF-based materials in animals larger than rodents, on which the most common applications of this material are drawn. An interesting study from 2021 was specifically aimed at the clinical use of artificial vascular grafts made from SF in dogs. The study showed that SF artificial vascular grafts implanted in the femoral arteries had satisfactory performance [68], but this is only one of the very few reports of tests conducted on companion animals. Hence, the valid claim in some publications that more studies on this topic are still needed [49].

## 7. Summary and Conclusions

In the current paper, the features of silk as material to be used in human and veterinary medicine has been characterised. It has been shown that silk, being easy and not expensive to obtain, is an excellent transplant material, because it fulfils main requirements for such materials, namely it has excellent cytocompatibility, is biocompatible, biodegradable and has low immunogenicity. Additionally, it has excellent mechanical properties, such as high ductility and breaking strength, even under extreme conditions (e.g., high temperature needed for sterilising). On the other hand, silk as such does not have bioactive properties, but its structure simplifies the enrichment with substances that can give it such properties.

SF scaffolds of various forms, such as mats, films, hydrogels, fibres, sponges or plastics for 3D printing, can be successfully applied in regenerative therapy of cartilaginous tissue. In addition, a method has been developed that offers the possibility of using SF to repair cartilage injuries using patient adipose tissue-derived stem cells stimulated to differentiate into chondrocytes.

Silk is also a polymer ideally suited to production of the nanostructures required for bone tissue regeneration. The electrospinning method used in this case allows the creation of fibres with various spatial orientations and the design of the mechanical properties of the resulting scaffolds. Hydrogels with silk are another form of bone damage treatment, where the presence of silk accelerated the therapeutic effect by increasing the rate of mineral deposition.

Although silk has been used for centuries to treat skin damage, modern research is focusing on silk-based 3D structures such as nanofibrous mats and microporous scaffolds. These structures must be combined with specific substances that confer bioactive properties: antibiotics, anti-inflammatory agents or agents that promote cell proliferation and migration. Dressings based on such enriched silk have a positive effect on accelerating the healing process of even the most challenging wounds.

SF-based hydrogels can be used to reconstruct small-diameter blood vessels. However, the hydrogel composition must be optimised for thrombogenicity and patency index, and then such materials can be ideally suited for coronary heart vessel transplants and by-passes.

The advantages of silk combinations with selected proteins and artificial polymers in regeneration of various tissues and organs are summarised in Table 1.

In conclusion, silk, which has historically held an undisputed position in the textile industry, can now find many other applications. Its biocompatibility, strength and versatility make silk a valuable material for a variety of medical applications, such as implantology, tissue engineering, wound dressings and drug delivery, all of which are discussed in this article. However, research into potential new applications and innovations in various medical fields, such as ophthalmology, neurology and dentistry, is ongoing.

Future research should be focused on using silk fibroin not only by itself as a scaffold, but mostly combined with other polymers and compounds, which will give it the properties of a bioactive material, as it may lead to excellent clinical results. Thus, using composite SF scaffolds can enhance cartilage, bone, skin and blood vessel regeneration, which would improve life of both veterinary and human patients that are now in need of invasive surgical procedures due to various tissue damage conditions.

## Figures and Tables

**Figure 1 materials-16-07128-f001:**
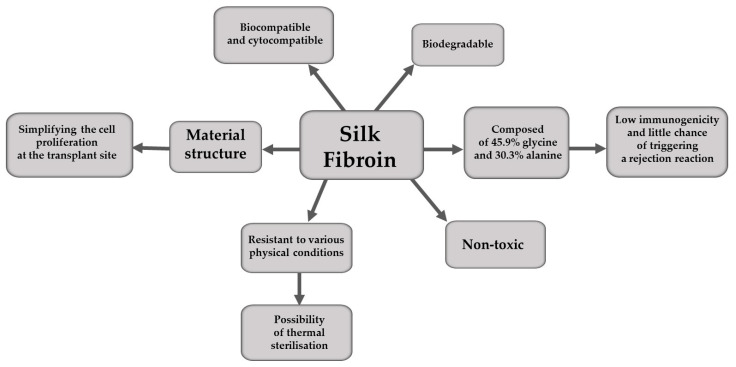
The advantages of silk fibroin.

**Figure 2 materials-16-07128-f002:**
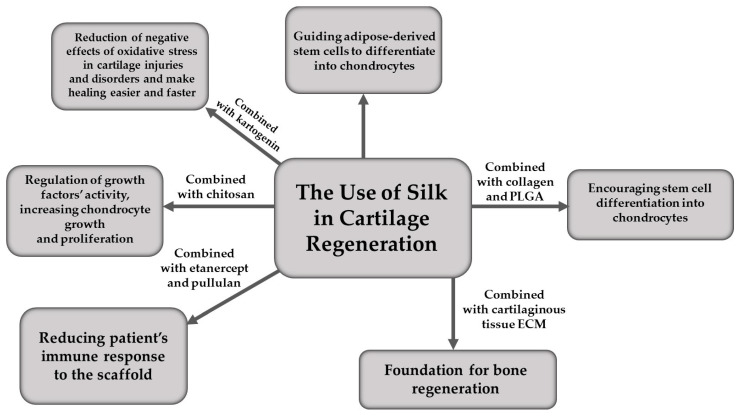
The use of silk in cartilage regeneration.

**Figure 3 materials-16-07128-f003:**
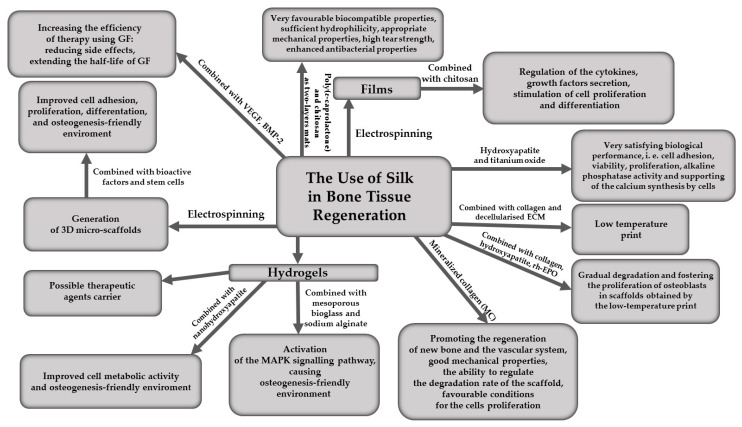
The use of silk in bone tissue regeneration.

**Figure 4 materials-16-07128-f004:**
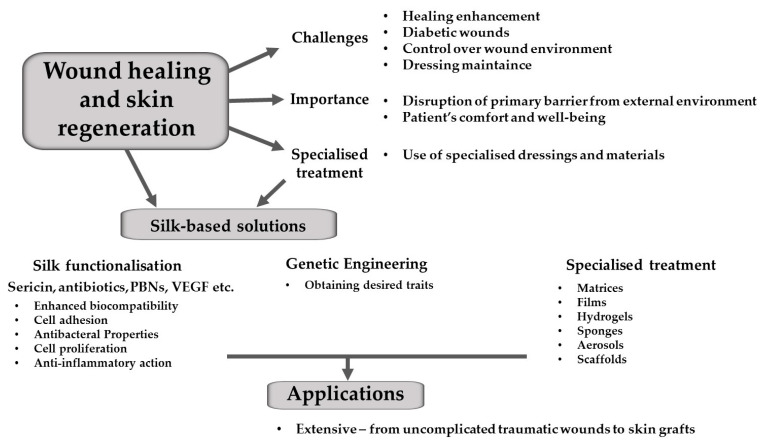
The use of silk in wound healing and skin regeneration.

**Figure 5 materials-16-07128-f005:**
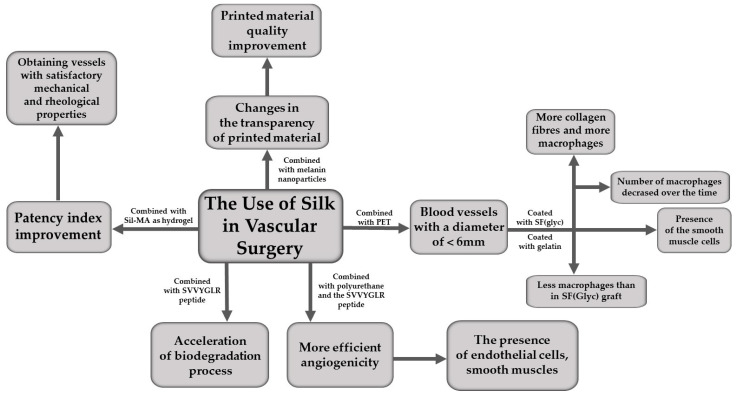
The use of silk in vascular surgery.

**Table 1 materials-16-07128-t001:** Advantages of silk combinations with selected proteins and artificial polymers in regeneration of various tissues and organs—summary.

Tissue/Organ	Protein/Polymer	Advantages
cartilage	chitosan	regulation of growth factors activity and chondrocytes proliferation [29,31]
	pullulan and etanercept	reduction of potential inflammatory reaction [36,38]
	poly(lactic-co-glycolic acid) microspheres and collagen	creating suitable environment for BMSCs differentiation into chondrocytes [37,39]
	kartogenin	reduction of negative effects of oxidative stress in cartilage injuries and disorders, and enhancement of healing process [33,34,35]
bone	PHBV	good conditions for cell adhesion, proliferation and differentiation [50]
	hydroxyapatite (HA)	induction of ECM production [50]
	HA as hydrogel	improvement of cell metabolic activity and bone regeneration efficiency [49,50,54]
	mesoporous bioglass and sodium alginate as hydrogel	activation of the MAPK signalling pathway, creating osteogenesis-friendly environment [57]
	growth factors (GF): VEGF, BMP-2	therapy efficiency improvement using GF: reducing side effects, extending the half-life of GF [55,56]
	HA and TiO_2_	satisfying biological performance, i.e., cell adhesion, viability, proliferation, alkaline phosphatase activity, and supporting of the calcium synthesis by cells [51]
	mineralised collagen (MC)	promotion of new bone and vascular system regeneration, good mechanical properties and ability to regulate the scaffold degradation rate, favourable conditions for cells proliferation [52]
	collagen and decellularised ECM	good environment for bone cells proliferation obtained by low temperature print [41,43]
	collagen, HA and recombinant human erythropoietin	gradual degradation and fostering of the osteoblasts proliferation in scaffolds obtained by the low-temperature print [20]
	poly(ε-caprolactone) and chitosan as two-layers mats	enhanced biocompatible and antibacterial properties, sufficient hydrophilicity, high tear strength and other mechanical properties improved [59]
	chitosan as film	regulation of cytokine and GF secretion and stimulation of cell proliferation and differentiation [58]
skin	sericin	enhancement of cell proliferation and antimicrobial properties [29]
	antibiotics (e.g., polymyxin)	limit of microbial activity [12]
blood vessels	Sil-MA as hydrogel	high patency rate of small diameter vessels [43]
	melanin nanoparticles	improvement of printed material quality [43]
	PET	promising material for grafts for small diameter blood vessels [68]
	polyurethane and SVVYGLR peptide	angiogenic-promoting activity [69]

## Data Availability

Not applicable.

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
