# Peer review of "Applications of Silk Fibroin in Human and Veterinary Medicine"

_materials, 2023, doi:10.3390/ma16227128_

Round 1
Reviewer 1 Report
Comments and Suggestions for Authors
This review paper explores the utilization of silk in conjunction with specific proteins and synthetic polymers in both human and veterinary medicine. The content of the review is compelling, but there are a few aspects that require attention before considering it for publication.
- The introduction of silk is somewhat perplexing. The authors initially delve into silk but then abruptly transition to discussing various types of silk fibers. This shift raises questions about whether these two concepts are synonymous or distinct.
- The clarity of the letters in Figure 2-6 could be improved.
- The referenced literature primarily consists of older sources, and there is an absence of references from the year 2023, which might impact the comprehensiveness of the review.
- The challenges associated with using silk as a biomaterial appear to be overlooked and should be addressed in the review to provide a more comprehensive perspective.
Author Response
Please, see the attachment.

Reviewer 2 Report
Comments and Suggestions for Authors
The author introduced the application of silk fibroin in cartilage, bone, skin and blood vessel tissue
engineering, and focused on the advantages of hydrogels and fiber containing structures, such as
electrospinning scaffolds. However, this manuscript only listssome studies without a clear structure,
and the theme of this manuscript is not clear enough. The current manuscript cannot present the
author's statement in the abstract part that "Therefore, silk is increasingly being used as a 3D printing
material in biomedicine." In addition, silk fibroin is a commonly used biomaterial, and there have
been numerous reviews of its application in tissue engineering. The organizational form and content
of this manuscript are similar to those of previous reviews, without highlighting new breakthroughs
in recent years, which is very important. I believe the manuscript requires further attention to detail
and do not recommend its publication in the current form.
In addition, there are some issues that need to be addressed:
1. Line78, what does “silk porosity”refer to?
2. Figure 2 - Figure 6 are very blurry, please provide high-definition images. These figures are
summary block diagrams, please provide references.
3. The functions mentioned in Figure 3, such as reducing immune response and regulating the
activity of growth factors, are not reflected in the text.
4. Why does the chapter on bone regeneration talk about blood vessels in Line168-173? It looks
very chaotic.
5. Line257-265, the author mentioned genetic engineering, which was very abrupt. Please explain
its association with other parts.
6. Table 1, Please provide references.

Author Response
Please, see the attachment.

Reviewer 3 Report
Comments and Suggestions for Authors
In this manuscript, Koczon et. al reviewed the use of silk-based materials in 4 different biomedical applications. In its current form, I think it is not suitable for publication in Materials journal. Substantial improvement of the work is necessary in my opinion for consideration. Below I am listing my points for this improvement.
1) I found the title of the review a bit odd, it is said that selected proteins and artificial polymers, however, in the text, there is not any classification regarding the other components of the biomaterials. Then why not just write Medical applications of Silk-based materials?
2) Abstract of the review needs to be rewritten, the info there suits introduction.
3) Introduction title is peculiar too. Why is it written as introduction-silk, as if there are other parts of introduction?
4) In my opinion, Figure 1 is horrible as whole, I would remove it completely. Instead, I would add a high quality scheme, summarizing the biomedical applications of silk-based materials.
5) Rest of the figures are difficult to read, and irritate eyes. And all of them contain just text, visually those figures needs to be improved substantially to attract readers. In its current form those figures looks like taken from a high school homework.
6) In Figure 2, there is an illogical statement. Why porous material is an advantage of silk fibroin? you can make almost all sorts of polymers porous? Material does not need to be porous to be enriched with additional substances? These advantages need to be rewritten in a more scientific way.
7) There are many other reviews about silk-based materials. In the introduction part, authors need to mention their motivation to prepare this work. What this work offers new? For example, we are almost at the end of 2023, but there is not a single citation from 2023.
8) Authors need to include at least couple of high-quality figures.
9) There should be future prospects section at the end.
10) Authors should enlarge the introduction part, and give some additional fundamental information regarding silk and biomaterials in general, and consider adding some extra citations for example,
Self-assembled silk fibroin hydrogels: from preparation to biomedical applications, Mater. Adv., 2022,3, 6920-6949
Preparation, properties, and applications of gelatin-based hydrogels (GHs) in the environmental, technological, and biomedical sectors. International Journal of Biological Macromolecules Volume 218, 1 October 2022, Pages 601-633
11) Language and grammer of the manuscript needs to be improved.
Comments on the Quality of English Language
It needs to be improved
Author Response
Please, see the attachment.

Round 2
Reviewer 1 Report
Comments and Suggestions for Authors
Accept
Reviewer 2 Report
Comments and Suggestions for Authors
The paper has been adequately revised. I recommend publication now.
Reviewer 3 Report
Comments and Suggestions for Authors
Authors have addressed my concerns
Comments on the Quality of English LanguageIt is ok